# Insights into Autotrophic Activities and Carbon Flow in Iron-Rich Pelagic Aggregates (Iron Snow)

**DOI:** 10.3390/microorganisms9071368

**Published:** 2021-06-23

**Authors:** Qianqian Li, Rebecca E. Cooper, Carl-Eric Wegner, Martin Taubert, Nico Jehmlich, Martin von Bergen, Kirsten Küsel

**Affiliations:** 1Institute of Biodiversity, Friedrich Schiller University Jena, Dornburger Strasse 159, 07743 Jena, Germany; qianqian.li@uni-jena.de (Q.L.); rebecca.cooper@uni-jena.de (R.E.C.); carl-eric.wegner@uni-jena.de (C.-E.W.); martin.taubert@uni-jena.de (M.T.); 2Department of Molecular Systems Biology, Helmholtz Centre for Environmental Research—UFZ, Permoserstrasse 15, 04318 Leipzig, Germany; nico.jehmlich@ufz.de (N.J.); martin.vonbergen@ufz.de (M.v.B.); 3Pharmacy and Psychology, Faculty of Biosciences, Institute of Biochemistry, University of Leipzig, Brüderstraße 32, 04103 Leipzig, Germany; 4The German Centre for Integrative Biodiversity Research (iDiv) Halle-Jena-Leipzig, Puschstraße 4, 04103 Leipzig, Germany

**Keywords:** iron snow, autotrophic iron oxidizing bacteria, heterotrophic iron reducing bacteria, carbon flow, metatranscriptomics, ^13^CO_2_, stable isotope probing, metaproteomics

## Abstract

Pelagic aggregates function as biological carbon pumps for transporting fixed organic carbon to sediments. In iron-rich (ferruginous) lakes, photoferrotrophic and chemolithoautotrophic bacteria contribute to CO_2_ fixation by oxidizing reduced iron, leading to the formation of iron-rich pelagic aggregates (iron snow). The significance of iron oxidizers in carbon fixation, their general role in iron snow functioning and the flow of carbon within iron snow is still unclear. Here, we combined a two-year metatranscriptome analysis of iron snow collected from an acidic lake with protein-based stable isotope probing to determine general metabolic activities and to trace ^13^CO_2_ incorporation in iron snow over time under oxic and anoxic conditions. mRNA-derived metatranscriptome of iron snow identified four key players (*Leptospirillum*, *Ferrovum*, *Acidithrix*, *Acidiphilium*) with relative abundances (59.6–85.7%) encoding ecologically relevant pathways, including carbon fixation and polysaccharide biosynthesis. No transcriptional activity for carbon fixation from archaea or eukaryotes was detected. ^13^CO_2_ incorporation studies identified active chemolithoautotroph *Ferrovum* under both conditions. Only 1.0–5.3% relative ^13^C abundances were found in heterotrophic *Acidiphilium* and *Acidocella* under oxic conditions. These data show that iron oxidizers play an important role in CO_2_ fixation, but the majority of fixed C will be directly transported to the sediment without feeding heterotrophs in the water column in acidic ferruginous lakes.

## 1. Introduction

Pelagic aggregates, formed in the water column of lakes and oceans through adsorption of inorganic and organic matter (OM), are composed of microorganisms, phytoplankton, feces, detritus and biominerals [1,2]. These snow-like aggregates are usually larger than 500 µm and are held together by extracellular polymeric substances (EPS) [3]. Marine pelagic aggregates (marine snow) drive the biological carbon pump via the export of photosynthetically derived particulate organic carbon (POC) from the photic zone of the ocean to the deep aphotic zones, where carbon can be sequestered for years before reaching the sediment [4,5,6]. The long residence time of marine snow enables microbial degradation and zooplankton grazing activity, which are largely responsible for the attenuation of carbon flux to the deep sea [7].

In iron-rich ferruginous meromictic lakes, iron-rich pelagic aggregates (iron snow) are formed and dominated by an Fe(III)-rich fraction of more than 35%, rather than OM, which speeds up their sinking velocity [8]. The world’s largest ferruginous lakes, such as the 600 m deep Lake Matano or Lake La Cruz, serve as analogous representatives to conditions of the Archaean Ocean ecosystems [9]. Here, low productivity in surface water allows sunlight to penetrate to the chemocline, stimulating anoxygenic phototrophic Fe(II)-oxidizing bacteria (FeOB) and anoxygenic green sulfur bacteria to form iron-rich pelagic aggregates [10,11,12,13]. In shallow ferruginous lakes, such as the numerous lignite lakes that have emerged in Europe during the last decades [14,15], Fe(II) was or still is provided by groundwater or surface water influx from mine tailings. At low pH conditions, Fe(II) oxidation is mediated mainly by chemolithotrophic acidophilic microbes [16]. The adsorption of phosphorous to the Fe(III) minerals formed within the water column limits primary production by phytoplankton in these lakes [17,18]. These limitations on primary production by phytoplankton suggest that carbon fixation mediated by FeOB might be central for primary production, further trophic interactions within iron snow and carbon export to the sediments.

In this study, we sampled iron snow from the model lignite Lake 77 (Lusatian district of Brandenburg, Germany), which also has a meromictic basin [8]. RNA-based quantitative PCR assigned up to 60% of the metabolically active iron snow colonizing microbes to Fe(II)-oxidizing and Fe(III)-reducing bacteria (FeRB) [8,19]. Fe-cycling key players, the FeOB *Acidithrix* and the FeRB *Acidiphilium*, isolated from iron snow [20], showed inter-species aggregation controlled by chemical signaling [21]. The heterotrophic *Acidiphilium* often co-occurs with the chemolithoautotrophic, EPS-producing FeOB *Ferrovum,* another iron snow key player [22,23].

To capture metabolic activities comprehensively and to follow the flow of carbon within iron snow, we used metatranscriptomics and a ^13^CO_2_ metabolic labeling approach. We collected iron snow below the redoxcline of lignite Lake 77 in two consecutive years (2017 and 2018) to profile the activities of the iron snow microbial community. Mapping mRNA sequences to the genomes of representative Fe-cycling bacteria helped to obtain gene expression profiling. Next, we incubated iron snow with ^13^C-CO_2_ under oxic and anoxic conditions and applied protein-based stable isotope probing (protein-SIP) over time to follow the labeling of peptides in the identified microbial community members. We hypothesized that dominant FeOB are mainly responsible for CO_2_ fixation and polysaccharide biosynthesis and that ^13^C-labeled organic carbon is rapidly utilized by the heterotrophic FeRB *Acidiphilium* and *Acidocella*, as well as other trophic levels. However, our results suggest the majority of fixed CO_2_ in this system is not feeding other trophic levels in iron snow; instead, the majority of the fixed CO_2_ is pumped into the sediment. 

## 2. Materials and Methods

### 2.1. Lake Characteristics and Iron Snow Sampling 

Lignite Lake 77 is an acidic coal mining lake located in the Lusatian mining area in eastern Germany. Iron snow sampling was conducted in both August 2017 and August 2018 at the central basin (CB, 51^°^1′8.2″ N, 13^°^41′34.7″ E) during lake stratification or at the very beginning of the ‘mixing’ period [8]. For lake samples used for Fe(II) and sulfate measurements, 1 mL of 5 M HCl was added to each 5 mL water sample collected from 0–6.5 m depth to prevent abiotic iron oxidation. Iron snow was collected directly onto glass fiber filters (0.9 µm; Infiltec, Speyer, Germany) using an electronic water pump installed between 5 and 6 m depth just below the redoxcline. An volume of 150 L of water was collected in 2017 and 50 L of water were collected in 2018. Glass fiber filters with iron snow were immediately stored on dry ice, transported to the laboratory and stored at −80 °C until nucleic acid extraction. For iron snow incubation experiments, additional lake water was collected (20 L of lake water from around 1 m depth and 100 L of water from 5–6 m depth) during the 2018 sampling campaign and stored in 20 L plastic bottles at 4 °C until further use. 

### 2.2. Geochemical Analysis

Temperature, pH, conductivity and dissolved oxygen content were measured with a multiparameter meter (YSI, Yellow Springs, OH, USA). Fe(II) concentrations were determined spectrophotometrically at 512 nm (Hach Lange, Düsseldorf, Germany) using the phenanthroline method [24]. Sulfate was measured spectrophotometrically at 420 nm, according to the barium chloride–gelatin method [25]. 

### 2.3. Nucleic Acid Extraction of Iron Snow Microbiome 

A single glass fiber filter with collected iron snow was cut into 12 even pieces, from which two even portions of filter pieces were pooled into 650 mL conical tubes and served as biological replicates. Then, 40–50 mL of oxalate extraction buffer (197 mM ammonium oxalate, 119 mM oxalic acid dissolved in DEPC-treated water, pH 3.25) were added to tubes and incubated for 30 min at 4 °C to dissolve the ferric iron minerals during which the tubes were shaken 3–5 times to detach cells from filters. Next, the oxalate buffer solution was passed through a 0.2 µm membrane filter (PALL corporation, Dreieich, Germany). Nucleic acids were extracted from these glass fiber membrane filters according to the method described in Lueders, Manefield and Friedrich 2004, with slight modification. First, the biomass-carrying membrane filters were washed in 15 mL conical tubes with 2.25 mL of NaPO_4_ buffer (120 mM, pH = 8) and 0.75 mL of TNS solution (500 mM Tris-HCl pH 8.0, 100 mM NaCl, 10% SDS *w*/*v*) and subsequently subjected to beadbeating at 6.5 m/s for 30 s with 0.6 g zirconia/silica beads (⌀ = 1 mm) (Carl Roth, Karlsruhe, Germany) to facilitate detachment. Cell debris was separated by centrifugation (20,000× *g* at 4 °C for 10 min) and the supernatant was first extracted with equal volumes of phenol-chloroform-isoamyl alcohol (25:24:1 *v*/*v*/*v*, AppliChem, Darmstadt, Germany), following centrifugation (20,000× *g* at 4 °C for 20 min), and second with chloroform-isoamyl alcohol (24:1 *v*/*v*, AppliChem, Darmstadt, Germany), following centrifugation (20,000× *g* at 4 °C for 20 min). The obtained aqueous phase was subjected to overnight precipitation at 4 °C with two volumes of polyethylene glycol 6000 (Carl Roth, Karlsruhe, Germany) and 5 μL of Glycogen (20 mg mL^−1^, Sigma-Aldrich, Darmstadt, Germany). Total nucleic acids were collected by centrifugation (20,000× *g* at 4 °C for 90 min). The resulting pellets were washed with 1.5 mL of ice-cold ethanol (70% *v*/*v*), following centrifugation (20,000× *g* at 4 °C for 30 min), and resuspended in 50 µL of elution buffer (Qiagen, Hilden, Germany). Of the resulting 6 total nucleic acid replicates, 3 were used for DNA-based amplicon sequencing and 3 replicates were used for metatranscriptome sequencing. 

### 2.4. 16S rRNA Gene Amplicon Sequencing

DNA concentrations were determined by a NanoDrop (Thermo Fisher Scientific, Waltham, MA, USA) and a fragment of the 16S rRNA gene was amplified using primers S-D-Bact-0341-b-S-17 (5′-CCTACGGGNGGCWGCAG-3′) and S-D-Bact-0785-a-A-21 (5′-GACTACHVGGGTATCTAATCC-3′) [26], using the following cycling conditions: 15 min initial denaturation, followed by 30 cycles of 45 s at 94 °C, 45 s at 55 °C, 45 s at 72 °C and a final extension at 72 °C for 10 min. PCR products were checked by agarose gel electrophoresis and the target gel bands were cut from gels and purified with the NucleoSpin gel and PCR clean-up kit (Macherey-Nagel, Düren, Germany). PCR products were prepared for Illumina sequencing using the NEBNext Ultra DNA Library Prep Kit for Illumina (New England Biolabs, Hitchin, UK), according to the manufacturer’s protocol. Sequencing was carried out in-house using an Illumina MiSeq platform (Illumina, San Diego, CA, USA) in paired-end mode (2 × 300 bp).

### 2.5. Amplicon Sequencing Data Analysis

Demultiplexing of raw data was performed with bclfastq (v. 2.19) (Illumina). Read pairs were imported into QIIME2 (v. 2019.7) [27] and primers were trimmed off by cutadapt (v. 2.5) with default parameters [28], which cuts the 5′ and 3′-end, taking into account respective primer sequences. Trimmed read pairs were subjected to paired-end assembly with vsearch (v. 2.7.0) (—fastq_mergepairs, default settings) [29] and then the joined sequences were filtered based on a minimum quality score of 30. Filtered sequences were denoised and amplicon sequence variants (ASVs) were identified with deblur (v. 2019.7) [30]. ASVs were taxonomically assigned against SILVA (release 132) [31] using the feature-classifier plugin of QIIME2 and a fitted classifier with the classify-sklearn (v. 0.21.2) function [32]. After rare ASVs with a frequency of less than 0.05% were removed, data were exported as biom file for downstream analysis with phyloseq (v. 1.28.0) [33] in R (v. 3.5.3) (R Core Team, 2020). 

### 2.6. Metatranscriptome Sequencing 

RNA was purified from the previously described nucleic acid extracts through enzymatic digestion with DNase (Thermofisher Scientific, Waltham, MA, USA) for 1 h at 37 °C. The digested RNA was checked by agarose gel electrophoresis and purified using the RNA Clean & Concentrator-5 kit (Zymo Research, Freiburg, Germany). Purity and quantity were determined by spectrophotometry and fluorometry. RNA-seq libraries were prepared using the NEBNext Ultra II Directional RNA Library Prep Kit for Illumina (New England Biolabs, Hitchin, UK) according to the manufacturer’s protocol. Libraries were pooled equimolarly and sequenced in paired-end (2 × 150 bp) mode on an Illumina HiSeq 2000 instrument by Eurofins Genomics (Constance, Germany). Between 51 and 110 million read pairs were obtained for each library. Data were processed according to Li et al. (2020). Briefly, the quality of the demultiplexed sequencing data was assessed using FastQC (v. 0.11.7) (http://www.bioinformatics.babraham.ac.uk/projects/fastqc/ (accessed on 10 January 2018)) followed by adapter trimming using trimgalore (v. 0.4.3, -q 20) (https://www.bioinformatics.babraham.ac.uk/projects/trim_galore/ (accessed on 25 January 2017)) and quality-filtering with sickle (v. 1.33, -q 20) (https://github.com/najoshi/sickle). The remaining read pairs were paired-end assembled with PEAR (v. 0.9.11) with default settings [34]. Sequences derived from mRNA were identified with SortMeRNA (v. 2.0) [35] with default settings using pre-compiled SILVA (release 132) [31] and Rfam databases (release 12.2) [36]. 

### 2.7. Taxonomic Annotation of rRNA-Derived Sequences

Subsets of 100,000 SSU rRNA-derived sequences were queried against NCBI RefSeq rRNA sequences (release 203) [37] using blastn (v. 2.9.0) [38] applying an e-value cutoff of 1 × 10^−3^ and collecting up to 10 database hits. The search output was then parsed in MEGAN (v. 6.17.0) [39] using the implemented lowest common ancestor (LCA) algorithm with default settings. Taxonomically assigned sequences were used to deduce relative abundances. Sequences of eukaryotic origin were extracted and imported into *QIIME2* (v. 2019.7) [27] for taxonomic analysis. 18S rRNA sequences were taxonomically classified against the SILVA database (release: 132) [31] using the classify-consensus-vsearch function of the feature-classifier plugin (v. 2019.7.0) of QIIME2 with an identity threshold of 95%. Data were exported as biom file, which was used for downstream analysis with phyloseq (v. 1.28.0) [33]. 

### 2.8. Taxonomic and Functional Profiling of mRNA Sequences

mRNA sequences were queried against NCBI RefSeq (release 203) [37] by diamond (v. 0.9.26.127) [40] in blastn mode applying an e-value cutoff of 1 × 10^−3^ and collecting the top database hit. The search output was imported into MEGAN (v. 6.17.0) [39] for taxonomic classification.

mRNA sequences belonging to bacteria were queried against the SwissProt database (release 2020_05) [41] with diamond (v. 0.9.26.127) [40] in blastx mode applying an e-value cutoff of 1 × 10^−3^ and collecting the top database hit. Sequences were annotated using the KEGG Orthology (KO) based on a custom mapping file that contains KO assignments for SwissProt sequences. KO assignments for SwissProt sequences were obtained using GhostKoala (v. 2.2) with default settings [42]. The numbers of mRNA sequences assigned to individual KOs were used for deducing relative abundance information about mRNA sequences linked to metabolic pathways of interest. 

### 2.9. Taxonomic Annotation of Functional Subsets

Sequences linked to functional categories of interest were extracted based on their KO assignment. These sequence subsets were subsequently queried against NCBI Refseq and taxonomically annotated using diamond and MEGAN as outlined above.

### 2.10. Metatranscriptome Read Recruitment Analysis of Dominant Fe-Cycling Microbes

All available genomes of highly abundant Fe-cycling microbes (*Leptospirillum* spp., *Acidithrix* spp., *Ferrovum* spp., *Acidiphilium* spp., *Acidocella* spp., *Granulicella* spp.) were downloaded as nucleotide fasta files from the NCBI assembly database (accessions are given in Appendix A). Anvi’o (v. 6.1) [43] was used to identify open reading frames with *prodigal* (v. 2.6.3) [44]. Identified genes were annotated based on blastp (v. 2.5.0), hmmer (v. 3.2.1) and KofamScan (v. 2020-03-02) [45,46,47] searches against the clusters of orthologous groups (COGs) database (release 2014) [48], Pfam (v. 32.0) [49] and KEGG (release 93.0) [50]. Metatranscriptome sequences derived from mRNA were mapped onto the collected genomes with bowtie2 (v. 2.3.5) applying default settings [51]. The resulting sam files were converted to bam files using SAMtools (v. 1.9) [52] with default settings. Read count tables were generated using featurecounts (v. 1.6.4) [53] and gene expression analysis was performed using the package edge R (v. 3.26.8) [54]. Genes with log_2_ counts per million (log_2_CPM) were plotted as distribution plot and log_2_CPM with the lowest values were set as cutoff values (Appendix A). Expressed genes were defined as genes featuring log_2_CPM values above the defined cutoff in at least three samples. Determined KO assignments of the expressed genes were used to profile the expressed metabolic potential. The coverage ratios of pathways were calculated by dividing the number of expressed genes by the number of genes in individual bacterial genomes corresponding to the individual pathway. 

### 2.11. Oxic and Anoxic Microcosm Incubation Experiments 

A volume of 120 L of lake water (20 L of lake water from around 1 m depth and 100 L of water from 5–6 m depth) was stored overnight at 4 °C to allow precipitation of iron snow. The lake water was then centrifuged at 702× *g* for 5 min at 4 °C and the iron snow-free lake water supernatant was separated from the iron snow precipitate. A total of 36 sterile 150 mL incubation serum bottles were set up with 1.1 mL of the collected iron snow and 120 mL of iron snow-free lake water each. FeSO_4_ was added to each replicate microcosm to a final concentration of 3 mM and sealed with butyl rubber stoppers. In oxic microcosms, oxygen sensor spots (PreSens, Regensburg, Germany) were adhered to the wall of each bottle to monitor O_2_ concentrations in the liquid phase. Anoxic microcosm incubations were flushed for two hours with sterile N_2_ (Linde, Pullach, Germany). Lastly, 4.2 mL of either ^13^C-labeled CO_2_ or unlabeled ^12^CO_2_ were added to each oxic and anoxic replicate. The microcosms were incubated in the dark at 15 °C for 28 days to induce chemoautotrophic CO_2_ fixation. During incubation, Fe(II) concentrations were determined as described above. Additional FeSO_4_ was added to oxic incubations to a final concentration of 3 mM, when measured concentrations reached 0 mM. The same incubations were spiked with additional 4.5 mL of 100% sterile O_2_ (Linde, Pullach, Germany) to yield a final concentration of ~250 μM, when the O_2_ concentrations decreased to 1/3 of the initial concentration. A Student’s t-test was used to compare the difference of microcosms under oxic and anoxic conditions and determine statistical significance (*p* < 0.05).

### 2.12. Nucleic Acid and Protein Extraction of Microcosms 

Sets of triplicate microcosms from ^12^CO_2_ and ^13^CO_2_ incubations under both oxic and anoxic conditions were harvested after 7, 14 and 28 days. The content of the microcosms was centrifuged at 1500× *g* for 5 min to separate iron snow from lake water. After that, iron snow pellets were washed with an oxalate buffer and incubated for 5–10 min until the brownish ferric iron minerals were completely dissolved. Cells were then pelleted by centrifugation at 22,000× *g* for 5 min at 4 °C, followed by transfer into 2.6 mL of SET buffer (0.75 M sucrose, 40 mM EDTA, 50 mM Tris base pH 9) in 15 mL conical tubes. Cell lysis was conducted by adding 350 μL 10% SDS and 20 μL 100 mM phenylmethylsulfonyl fluoride (PMSF) in 2-propanol and subsequent incubation for 2 h at 60 °C, 800 rpm (HTA-BioTec, Bovenden, Germany). The obtained solution was extracted twice with 2 mL of phenol-chloroform-isoamyl alcohol (25:24:1 *v*/*v*/*v*, AppliChem, Darmstadt, Germany) and twice with 2 mL of chloroform-isoamyl alcohol (24:1 *v*/*v*, AppliChem, Darmstadt, Germany), centrifuging for 5 min at 3220× *g*. The obtained aqueous phase was subjected to overnight precipitation at −20 °C with 5 μL of glycogen (20 mg mL^−1^, Sigma-Aldrich, Darmstadt, Germany), 1 mL of NH_4_OAc and 8 mL of pure ethanol. Total nucleic acids were collected by centrifugation at 3220× *g* and 4 °C for 30 min. The resulting pellets were washed with ice-cold ethanol (80% *v*/*v*), centrifuged at 16,260× *g* at 4 °C for 20 min and suspended in 100 µL of TE buffer.

The phenol phase was collected in 15 mL falcon tubes, washed with 1 mL of SET buffer and centrifuged at 3220× *g* for 10 min at 4 °C. The bottom phenol phase was transferred to a 50 mL falcon tube containing 20 mL of 0.1 M NH_4_OAc (dissolved in MeOH), precipitated overnight at −20°C and pelleted by centrifugation (3220× *g* at 4 °C for 1 h). The protein pellets were washed twice with 0.1 M NH_4_OAc, twice with 80% acetone and once with 70% EtOH. Each washing step of pellets was followed by incubation (−20 °C for 20 min) and centrifugation (16,260× *g*, 4 °C for 10 min.). The resulting protein pellets were air dried for 15 min and stored at −20°C.

### 2.13. Metaproteomics Analysis: Identification, Taxonomy, Function and ^13^C Quantification of Peptides from Proteins

Protein extracts were subjected to one-dimensional SDS polyacrylamide gel electrophoresis, followed by tryptic digestion using Trypsin Gold (Promega, Madison, WI, USA), as previously described [55]. The resulting tryptic peptides were desalted and enriched using ZipTip-μC18 (Merck, Darmstadt, Germany). The obtained peptides were resuspended in 0.1% formic acid and analyzed on a Q Exactive HF instrument (Thermo Fisher Scientific, Waltham, MA, USA), equipped with a TriVersa NanoMate source (Advion Ltd., Ithaca, NY, USA) in LC chip coupling mode. Mass spectrometric raw data were analyzed using Proteome Discoverer (v1.4.1.14, Thermo Fisher Scientific, Waltham, MA, USA) with the Sequest HT search algorithm. Protein identification was performed using a self-built reference database based on microbial community composition of the microcosms determined by 16S rRNA gene and metatranscriptome analysis (see above). Protein sequences of respective microbial genera were downloaded from UniProt (release 2020_02) [56], or from NCBI, if no sequences from UniProt were available. Searches were conducted with the following parameters: Trypsin as enzyme specificity and two missed cleavages allowed. A peptide ion tolerance of 10 ppm and an MS/MS tolerance of 0.05 Da were used. As modifications, oxidation (methionine) and carbamidomethylation (cysteine) were selected. Peptides that scored a q-value >1% based on a decoy database and with a peptide rank of 1, were considered identified. Functional classification of identified proteins was conducted by mapping accession numbers from Uniprot to the KEGG Orthology (KO) database (Release 93.0) [50]. For proteins without mapped KO, amino acid sequences were classified using BlastKoala (v.2.2) [42]. 

The identification of ^13^C-labelled peptides and the calculation of their ^13^C relative incorporation abundance (RIA) were performed by comparing theoretical and experimental isotopologue patterns, chromatographic retention times and fragmentation patterns as previously described [57]. The RIA was used to distinguish between active autotrophs assimilating ^13^CO_2_ (RIA above 90%), active heterotrophs potentially labeled by crossfeeding (RIA between 1.1% and 90%) and inactive microbes (RIA of 1.1%, i.e., natural abundance of ^13^C). Generation times of *Leptospirillum*, *Ferrovum* and *Acidiphilum* were calculated based on the relative intensity of the mass spectrometric signals of unlabeled and labeled peptides as previously described [57]. Firstly, the number of doublings n was calculated using Equation (1), where $I_{12_{C}}$ is the signal intensity of the unlabeled peptide and $I_{13_{C}}$ is the signal intensity of the labeled peptide:(1)$$n=log_{2}\frac{I_{12_{C}}+I_{13_{C}}}{I_{12_{C}}}$$

In case of overlapping signals of labeled and unlabeled peptides, the monoisotopic peak was used to determine the total abundance of unlabeled peptide based on the natural distribution of heavy isotopes, as previously described [58]. Finally, the generation time $t_{d}$ was calculated using Equation (2), where Δt is the incubation time:(2)$$t_{d}=\frac{\Delta t}{n}$$

### 2.14. Statistics

Statistical analyses were performed with the R software framework (v. 3.5.3) (R Core Team, 2020). Permutational multivariate analysis of variance (PERMANOVA) was used to compare microbial communities between different years or different incubations. The Pearson correlation coefficient was calculated to determine if oxygen and iron consumption were correlating. All other testing was performed using a Student’s *t*-test. Test results were considered significant if two-tailed *p* values were smaller than 0.05.

### 2.15. Figure Generation

The UpSetR package [59] was used to create the intersect plot depicting overlapping KOs between the 2017 and 2018 samples. Unless otherwise stated, all figures were produced using the R package ggplot2 (version) (Wickham 2016) and polished with Inkscape (v. 0.92) (https://inkscape.org/ (accessed on 01 January 2017).

## 3. Results

### 3.1. Physicochemical Characterization and Microbial Community Composition and Diversity of Lake 77

The sampling at Lake 77 in August 2017 and 2018 was conducted prior to the beginning of the mixing period between surface water (epilimnion) and the anoxic hypolimnion fed by Fe(II)-rich groundwater. In 2017, the stratification was beginning to diminish due to the onset of mixing, resulting in a downward shift of the redoxcline (3.5–5 m), while the stratification and redoxcline (3–4.5 m) were more defined in 2018 (Figure 1A). In both years, profiles of opposing gradients of Fe(II) and O_2_ were observed with increasing depths. A decline in dissolved O_2_ was observed at the upper depths of the redoxcline and decreased to ~0 mg L^−1^ just below the redoxcline, while Fe(II) concentrations increased to 0.32 and 0.73 mmol L^−1^ in 2017 and 2018, respectively, below the redoxcline. The acidic pH (2.3–2.4 in 2017, 2.5–2.6 in 2018) and conductivity (2.9–3.2 mS cm^−1^ in 2017, 2.7–3.1 mS cm^−1^ in 2018) remained stable across depth, whereas the temperature decreased with increasing depth (from 22 to 11 °C). Sulfate concentrations fluctuated over depths, ranging 16.17–19.64 mM in 2017 and 15.84–19.34 mM in 2018 (Appendix A).

To profile the iron snow microbial community and to identify key players based on transcriptional activity, we performed 16S rRNA gene amplicon and total RNA sequencing. In addition, we were interested in investigating if outlined differences with respect to different sampling years were reflected in microbial community structure. Overall, the microbial compositions of the two different years were similar (PERMANOVA, *p* > 0.05) (Appendix A). Between 21.6 and 41.4% of SSU rRNA-derived metatranscriptome sequences were taxonomically assigned at the genus level (17.9%–37.2% of sequences linked to bacteria and 3.6–6.2% related to eukaryotes). Iron snow microbiome taxonomic profiles of 16S rRNA sequences showed the dominance of bacteria involved in Fe-cycling (72.6–92.2%) (Appendix A), while high relative abundances were detected for *Stramenopile* (74.6–91.7%) in 18S rRNA sequences (Appendix A). Both 16S rRNA gene amplicon sequencing and transcriptome sequencing showed that the microbial community was predominantly composed of FeOB (*Leptospirillum*, *Ferrovum*, *Acidithrix*) and FeRB (*Acidiphilium*, *Acidocella*) (Figure 1B,C, Appendix A). The relative abundance of *Leptospirillum* in 2017 was significantly higher than the relative abundance of *Leptospirillum* in 2018 (51.4% and 15.0%, respectively), whereas the relative abundance of *Acidiphilium* was higher in 2018 in comparison to 2017 (27.5% and 14.5%, respectively) based on mRNA sequences (Figure 1C). In addition to Fe-cycling microbes, we also detected polysaccharide-degrading bacteria, such as *Granulicella* (1.1–6.4%), and the phototroph, *Rhodopila* (1.8–6.6%), in the metatranscriptome datasets (Figure 1C). The phototrophic Cyanobacteria (0.04–0.08%) were only detected in the 2017 16S rRNA amplicon datasets.

### 3.2. Gene Expression Analysis of Iron Snow Microbial Communities 

The proportion of mRNA sequences ranged from 4.4% to 13.1% in the metatranscriptome datasets (Appendix A). Between 18.1% and 41.6% of the mRNA sequences were assigned to bacteria. Substantially smaller fractions were linked to eukaryotes (0.16–0.75%) and archaea (0.11–0.23%). In total, 25.7–30.7% of bacterial mRNA sequences were functionally assigned based on KEGG orthology (KO). In total, 3427 and 3436 different functions were identified based on KOs for the 2017 and 2018 metatranscriptome datasets, respectively (Figure 2A). A total of 91.0% of the KOs was shared between the two datasets. In order to identify direct links between taxonomy and functions, mRNA sequences of representative KOs/gene functions based on metabolic pathways of interest were subsequently taxonomically assigned (Appendix A). Genera classified as either FeOB (*Leptospirillum*, *Ferrovum* and *Acidithrix*) or FeRB (*Acidiphilium*) showed the highest transcriptional activity for functions linked to CO_2_ fixation, polysaccharide biosynthesis and motility (Figure 2B). 

We were able to assign bacterial mRNA sequences to the following complete CO_2_ fixation-related pathways: the Calvin–Benson–Bassham (CBB) cycle (2.0–2.8%) and the reverse tricarboxylic acid (rTCA) cycle (2.2–2.6%). The percentages are in relation to all other pathways. In total, 6.1–57.6% of sequences affiliated to key genes *rbcL* (ribulose-bisphosphate carboxylase large chain), *rbcS* (ribulose-bisphosphate carboxylase small chain), *frdABCD* (fumarate reductase gene cluster), *korABCD* (2-oxoglutarate/2-oxoacid ferredoxin oxidoreductase), *aclABY* (ATP-citrate lyase), *ccsAB* (citryl-CoA synthetase), *ccl* (citryl-CoA lyase) of the two CO_2_ fixation pathways mentioned above were assigned to *Leptospirillum.* However, few mRNA sequences of *Ferrovum* were mapped to the above key genes. We only detected a few mRNA sequences linked to archaea and eukaryotes and did not detect transcriptional activity for carbon fixation from archaea or eukaryotes. Additionally, there were 0.06–0.26% of sequences linked to photosynthetic reaction centers (Appendix A) and 98% of these sequences were mapped to *Acidiphilium*. *Acidiphilium rubrum* was previously found to show light-stimulated CO_2_ uptake [60,61]. Genes linked to anoxic photosystem II (*pufABLM*, *puhA*) were partially expressed. Of these mRNA sequences, only 1.08% were mapped to photosynthetic bacteria (*Rhodopila*, *Rhodopsudomonas*). 

We identified mRNA sequences of genes predicted to be involved in the metabolism of nucleotide sugar, such as UDP-glucose, UDP-galactose and UDP-glucuronic acid, all of which may serve as precursors for the synthesis of exopolysaccharides. Additionally, we mapped mRNA sequences to the genes involved in the exopolysaccharides biosynthesis, such as *cysE* (serine O-acetyltransferase), involved in vibrio polysaccharide biosynthesis in *Vibrio cholerae*, *rfbN* (rhamnosyltransferase), involved in Psl polysaccharide biosynthesis in *Pseudomonas aeruginosa*, and *wza* (polysaccharide biosynthesis/export) and *pgaBC* (poly-N-acetyl-glucosamine biosynthesis) in *Escherichia coli*. Gene functions linked to EPS production (nucleotide sugar metabolism, exopolysaccharide biosynthesis) were present in 2.6–3.8% of the mRNA sequences. The FeOB *Leptospirillum*, *Ferrovum* encoded 24.9–75.2% of the mRNA sequences linked to EPS production. 

We also mapped mRNA sequences to polysaccharide breakdown enzymes, such as β-glucosidase and endoglucanase. Bacterial gene functions linked to organic carbon utilization, for example glycolysis and polysaccharide breakdown, made up between 3.5% and 4.8% of the bacterial mRNA sequences. Of these mRNA sequences, 23.7–54.2% were assigned to the heterotrophs *Acidithrix* and *Acidiphilium*. Additionally, we found mRNA sequences mapped to genes linked to the Fe(II) oxidation (0.3–2.6%), of which 98.6% of sequences were mapped to *Leptospirillum*, *Ferrovum* and *Rhodopila* in both datasets. mRNA sequences linked to Fe(III) reduction were not detected, due to a lack of knowledge of Fe(III) reduction machineries in acidophiles. Motility was essential in the iron snow microbiome with assignment rates (1.4–2.6%) to chemotaxis and flagellar assembly, of which 68.8% of sequences were assigned to *Leptospirillum*, *Ferrovum*, *Acidithrix* and *Acidiphilium*. 

### 3.3. Taxonomic Profiles of Active FeOB and FeRB Taxa in Oxic and Anoxic Iron Snow Microcosm Incubations

To identify the active key players in the iron snow, we performed ^13^CO_2_ SIP combined with metaproteomics in iron snow microcosm incubations, amended with iron snow samples collected in 2018. The microcosms were incubated under oxic and anoxic conditions, similar to conditions above and below the redoxcline of Lake 77. In the oxic microcosms, the rate of oxygen consumption increased (Student’s *t*-test, *p* < 0.001) 15-fold, from 1.69 to 25.64 µM day^−1^, from 1–7 days (T1) to 15–28 days (T3). The rate of Fe(II) oxidation increased (Student’s t-test, *p* < 0.05) 2.6-fold from 0.56 to 1.48 mM day^−1^ in the same period (Appendix A). Oxygen consumption rates showed a highly significant correlation to iron consumption rates (Pearson correlation, *p* < 0.001) (Appendix A), demonstrating that active iron oxidizers were present in the microcosms. However, there were no significant changes in the Fe(II) concentration in anoxic microcosms. 

The dominant Fe-cycling bacteria (*Leptospirillum*, *Ferrovum*, *Acidithrix*, *Acidiphilium* and *Acidocella*) and the second most dominant bacteria (*Granulicella* and *Rhodopila*) in iron snow microcosms were consistent with the iron snow microbiome composition in the metatranscriptome datasets (Figure 3B). However, microbial compositions based on 16S rRNA amplicon sequencing and the relative abundances of peptides changed significantly between incubation conditions at three time points (PERMANOVA, *p* < 0.001) (Figure 3A, Appendix A). Consistent with the absence of Cyanobacteria based on 16S rRNA amplicon sequencing of in situ iron snow in 2018, we did not detect 16S rRNA sequences assigned to Cyanobacteria in either the oxic or anoxic microcosms. The relative abundances of the iron snow key player, *Acidithrix*, were significantly higher (Student’s *t*-test, *p* < 0.001) under anoxic conditions in comparison to oxic conditions, but the relative abundances did not differ over different incubation times. Among the top 10 bacteria which drive the difference between the microbial communities in the cluster of microcosms under oxic and anoxic conditions, the different abundances of FeOB rather than FeRB contributed the most differences of microbial communities (Appendix A). 

### 3.4. Carbon Flow of Dominant Iron Cycling Bacteria in Iron Snow Microcosms

Investigation of the C flux revealed *Ferrovum* and *Leptospirillum* fixed CO_2_ in the oxic microcosms, leading to a ^13^C relative isotope abundance (RIA) higher than 90% in their peptides (Figure 4A). These active chemolithoautotrophs showed generation times of 6 and 16 days within the first week of incubation, respectively (Appendix A). Peptides of *Acidocella* and *Acidiphilium* displayed a much lower ^13^C RIA of 1.6% at T1, increasing to 2.7%, and 2.8% at T3. These incorporation patterns suggest heterotrophic growth on a mostly unlabeled source of organic carbon in the iron snow, receiving minor input of ^13^CO_2_-derived carbon from the chemolithoautotrophs. Interestingly, the generation times of *Acidiphilium* (9–20 days) were in the same range as those of the chemolithoautotrophs, indicating fast growth. Surprisingly, *Candidatus* Finniella and *Spirochaeta* showed RIA greater than 90% at T3. Both of these organisms are strictly heterotrophic, with *Candidatus* Finniella being an endosymbiont of a protist [62] and *Spirochaeta* is heterotrophic utilizing different mono-, di- and oligosaccharides (e.g., pentose, starch) [63]. This suggests that these bacteria got labeled by cross-feeding of ^13^CO_2_-derived organic carbon directly from *Leptospirillum* and *Ferrovum* and not the unlabeled organic carbon source in the iron snow that *Acidiphilium* and *Acidocella* were using. Their generation times were 17 and 12 days at T3, separately. *Acidithrix* and *Granulicella* were not found to be active in the microcosms, although they were highly abundant in the in situ iron snow microbiome and oxic and anoxic microcosms. The larger deviation of the RIA for *Ferrovum* and *Leptospirillum* at T2 was caused by one replicate microcosm with a significantly lower RIA in the highly labeled organisms, likely due to a contamination with approximately 10% ^12^CO_2_a.

The generation time of *Ferrovum* under anoxic conditions with average 32 days was significantly higher (Student’s *t*-test, *p* < 0.01) than 6 days under oxic conditions within the time frame investigated, implying slower growth (Appendix A). In the anoxic microcosms, only *Ferrovum* fixed CO_2_, leading to RIA greater than 90%. None of the remaining bacteria showed detectable metabolic activity within the time frame investigated (Figure 4B).

### 3.5. Metatranscriptomic and Metaproteomic Functional Profile of Dominant Microbes

The dominance of Fe-cycling bacteria and carbon transfer between FeOB and FeRB suggests the Fe-cycling key microbes play an important role in iron snow. In order to profile the functions of the key players in the in situ microbiome below the redoxcline and in microcosms under oxic and anoxic conditions, we mapped mRNA sequences to the available *Leptospirillum*, *Ferrovum*, *Acidithrix*, *Acidiphilium*, *Acidocella* and *Granulicella* genomes in NCBI assembly database (Appendix A) and profiled peptides linked to these key players in both the oxic and anoxic microcosms (Figure 5). Among these mapped genomes, iron snow isolates *Acidithrix* sp. C25, *Acidiphilium* sp. C61, *Acidocella* sp. C78 recruited most of the mRNA sequences. In total, 84.4%, 84.1% and 92.2% of their genomes were found to be expressed, respectively (Appendix A). *Ferrovum* expressed the whole set of CBB cycle genes for CO_2_ fixation. While most of the rTCA cycle genes were expressed by *Leptospirillum*, we did not find any mRNA sequences mapped to the key genes in the *korABCD* (2-oxoglutarate/2-oxoacid ferredoxin oxidoreductase) and *aclABY* (ATP-citrate lyase) operons. In agreement, peptides associated with the CBB cycle and rTCA cycle were mostly affiliated with *Ferrovum* and *Leptospirillum* (Figure 5). Peptides linked to exopolysaccharide biosynthesis (e.g., *wza*) were found in chemolithoautotrophic *Leptospirillum* in oxic microcosms and *Ferrovum* in both oxic and anoxic microcosms. However, peptides linked to polysaccharide breakdown were identified in *Acidiphilium* in oxic microcosms, but not *Acidocella*, despite the identification of peptides linked to organic carbon utilization in *Acidiphilium* and *Acidocella*. We observed peptides linked to flagellar assembly in *Acidithrix*, *Acidiphilium* and *Acidocella* in oxic microcosms, but not in the anoxic microcosms.

## 4. Discussion

The acidic ferruginous model Lake 77 provides ideal conditions to investigate the extent to which Fe(II) oxidation linked autotrophy (chemolithoautotrophy) contributes to the biological carbon pump versus the contribution from classical photoautotrophy. The Fe-cycling microbes found in Lake 77 mimic the previously characterized AMD and pit lake microbial communities [64,65]. Our molecular data consistently show the dominance of autotrophic FeOB (*Leptospirillum* and *Ferrovum*) in iron snow, whereas autotrophic communities in marine or lake snow are dominated by phytoplankton (i.e., cyanobacteria and algae). Phytoplankton contributes to more than 50% of overall primary production in marine snow or lake snow [66,67,68]. The most abundant phototrophic bacterium, *Rhodopila*, accounted for 88.1–93.3% of allphototrophs detected in the iron snow of Lake 77. However, the relative abundances of mRNA sequences linked to *Rhodopila* ranged from 1.8 to 6.6%. Members of this genus are described as acidophilic anaerobic anoxygenic phototrophic purple bacteria [69]. After mapping mRNA sequences onto the *Rhodopila* genomes, we did not detect gene expression for all genes of the CBB-cycle. Transcripts linked to the small subunit of RuBisCO were not found. The mRNA sequences associated with eukaryotes were 31–388 times lower than the mRNA sequences mapped to bacteria in our samples. Several key genes of CO_2_ fixation pathways were missing in mRNA sequences mapped to eukaryotes, including the small and large chain RuBisCO genes and a fumarate reductase gene cluster. Based on these findings, photosynthetic CO_2_ fixation of eukaryotes and bacteria is of minor importance in iron snow, which further elucidates the difference in microbially mediated processes across environments characterized by the presence of pelagic aggregates. For example, the dominant bacteria in marine or lake snow are members of the Bacteroidetes phylum, including the *Cytophaga* and *Flavobacteria* genera, and the Proteobacteria phylum, including α-, β- and γ-Proteobacteria, all of which are mainly associated with degrading OM during sinking of these pelagic aggregates [70,71].

Despite the differences in microbial community composition between iron snow and marine or lake snow, the functional properties of the aggregate communities were rather similar. In addition to genetic pathways responsible for CO_2_ fixation, we also found evidence for other key microbially mediated processes found in marine snow, e.g., polysaccharide biosynthesis [72], organic carbon hydrolysis [73], nitrogen fixation [74], sulfate reduction [75] and motility [76]. The most abundant Fe-cycling microbes (*Leptospirillum*, *Ferrovum*, *Acidithrix* and *Acidiphilium*) comprise more than 50% of the total iron snow community. These four microbes also represent the community members with the highest transcriptional activity based on mRNA abundances. Transcripts linked to CO_2_ fixation, EPS biosynthesis, electron transfer processes, nitrogen fixation, sulfate reduction and flagellar-based motility were the most abundant. However, abundances of mRNA sequences involved in N_2_ fixation were low (0.01%–0.09%). At the species level, two of our iron snow isolates, *Acidithrix* sp. C25 and *Acidiphilium* sp. C61, similarly expressed the most genes linked to these general activities. The high abundances of mRNA sequences mapped to *Leptospirillum* and *Ferrovum* in our datasets were linked to CO2 fixation and polysaccharide production, suggesting that these chemolithoautotrophic FeOB are the key species also driving EPS production. Conversely, in marine snow, phytoplankton (e.g., diatoms, cyanobacteria) produce transparent exopolymeric particles (TEP) or exopolysaccharides, which mediate cell aggregation and coagulation [77,78,79]. Cyanobacteria are mainly responsible for the marine or lake snow dinitrogen (N_2_) fixation [80,81], with an annual contribution ranging from 26 to 47% in the North Pacific Subtropical Gyre [82], but do not play a significant role in iron snow. 

Due to pyrite oxidation in Lake 77, concentrations of sulfate are high in the water column (up to ~20 mM), however, mRNA sequences linked to assimilatory/dissimilatory sulfate reduction were relatively low, ranging from 0.40% to 0.58%. The sulfate concentration is 11–16× higher than in meromictic Lake Cadagno [83]. Microbially mediated sulfate-reduction leading to the production of sulfide in marine and lake snow occurs in lower anoxic zones of the water column [75,84]. Sulfide concentrations within marine snow range from 1.3 to 25 μmol S L^−1^ [75], while Lake Cadagno contains 1.5–1.9 mmol S L^−1^ in the monimolimnion [85]. We found evidence for the activity of sulfate reducers with extremely low relative abundance (~0.3%) in iron snow (e.g., *Desulfosporosinus*, *Desulfobulbus*). The low pH conditions in mining lakes limit sulfate reduction to sediment zones with higher pH [86,87], as only few sulfate reducers are known to tolerate low pH conditions [88,89].

Specific to iron snow microbiome activities, the chemical mediator-driven interactions observed between *Acidithrix* sp. C25 and *Acidiphilium* sp. C61 are in part due to the production and detection of PEA by these microbes, respectively [21]. We found 0.004–0.01% of mRNA sequences linked to amino-acid decarboxylase, which is linked to PEA production. PEA produced by *Acidithrix* sp. C25 induces aggregation in *Acidiphilium* strains [21,90]. Marine snow-associated bacteria produce acylated homoserine lactones (AHLs) to regulate microbial colonization and coordinated group behavior by quorum sensing (QS) [91,92]. We were able to map mRNA sequences to autoinducer-1 synthesis (i.e., acyl-homoserine lactose synthase AI-1) (up to 0.003%) in *Acidocella* spp. and receptor genes (i.e., LuxR transcriptional regulator, autoinducer sensor kinase) (up to 0.004%) in *Leptospirillum* spp., *Ferrovm* spp., *Acidiphilium* spp. and *Granulicella* spp. [91,92]. However, QS is not required for lake snow-attached bacteria. They can rely on novel signal molecules to mediate cell–cell signaling in lake snow [93]. Taken together, microbially mediated communication appears to be driven by chemical mediators in iron snow.

Similar to marine and lake snow-attached bacteria, which are known to extracellularly hydrolyze the phytoplankton-derived fixed carbon, the iron snow microbiome can utilize oligosaccharides produced via the extracellular hydrolysis of biological materials (i.e., fixed carbon, detritus) and function as common goods to promote cooperative growth. Since carbon sources are severely limited in acidic waters [94], obligate interactions between members of microbial consortia are probably critical in the optimization of microbial activity under acidic conditions [95]. To elucidate the importance of chemolithoautotrophic-mediated CO_2_ fixation for the whole iron snow microbial community, rates of incorporation and the carbon flow were elucidated in oxic and anoxic microcosm incubations. ^13^C RIA values above 90%, demonstrating chemolithoautotrophic growth, were detected in peptides of *Ferrovum* and *Leptospirillum.* Peptides specifically linked to the CBB pathway were predominantly affiliated with *Ferrovum*. The ^13^CO_2_ fixed by these taxa (i.e., the newly available organic carbon) could subsequently be transformed to polysaccharides and ultimately used for EPS production. We observed a slow increase in ^13^C quantification values in the heterotrophs *Acidiphilium* and *Acidocella*, but not *Granulicella*, in oxic microcosms, which suggested utilization of ^13^CO_2_-derived organic carbon by these most abundant heterotrophic community members. *Acidiphilium* species have the capacity to break down EPS [90,96] and couple Fe(III) reduction to oxidation of sugars [97,98]. In the metatranscriptome dataset, we found *Acidiphilium* mRNA sequences linked to polysaccharide-breakdown enzymes, such as β-glucosidases and endoglucanases, which hydrolyze polysaccharides to monosaccharides. Overall, these results suggest that CO_2_ fixed by the iron snow chemolithoautotrophic FeOB is converted to polysaccharides and EPS, which provide an organic C source to the heterotrophic members, as also described elsewhere [21]. In addition, EPS initiates the cohesiveness and contributes to the structural stability of iron snow [20]. 

In the anoxic microcosm incubations, only *Ferrovum* had measurable ^13^C incorporation. *Ferrovum* has the metabolic potential to oxidize Fe(II) and reduce NO_3_^−1^ [23,99] coupled to CO_2_ fixation, which further explains why its growth was not restricted to oxic conditions. In addition, we detected mRNA sequences linked to nitrate reductase in iron snow which was collected below the redoxcline in 2018, although we could not identify proteins linked to the nitrate reductase in anoxic microcosms. Surprisingly, FeRB did not show any activity with regards to measurable ^13^C incorporation, suggesting a limited flow of carbon to other trophic levels when iron snow is sinking through the anoxic hypolimnion of Lake 77. Instead, our functional profile analysis and corresponding identification of peptides linked to flagellar motility indicate motility is still essential for the colonization of EPS-stabilized iron snow aggregates, consistent with natural assemblages of marine bacteria that express specific flagellin genes and exhibit motility [100,101]. 

Modern ferruginous meromictic lakes are important analogs to study microbial processes involved in carbon cycling in the Archean and Proterozoic oceans [102], where primary production was likely driven by anoxygenic photosynthetic Fe(II) oxidizing bacteria (“photoferrotrophs”) [11,13]. In Lake Matano and Lake La Cruz, for example, phosphorus limitation controls primary production in the oxic layers of Lake Matano and Lake La Cruz [103,104], which allows light penetration below the oxic–anoxic interface. Here, sulfide-oxidizing phototrophic bacteria and anoxygenic photoferrotrophs drive photosynthetic CO_2_ fixation and regulate OM export [13,105,106]. Similarly, sorption of phosphorous to the high amounts of ferric iron negatively affects the eukaryotic primary activity in Lake 77 and similar environments [105]. Despite the presence of sulfur oxidation genes (*soxA* and *dsrAB*) and iron oxidation genes (*cyc2*), incomplete anoxic photosynthetic reaction center genes in photosynthetic bacteria suggested anoxic photosynthesis coupling sulfide/Fe(II) oxidation was unlikely to happen in iron snow microbiome below the redoxcline. As microcosms were incubated in the dark, we might have missed ^13^C incorporation in the most abundant anoxic photosynthetic bacteria, *Rhodopila.* Low ^13^C incorporation by heterotrophs after the 28-day incubation period under oxic conditions provides strong evidence that the majority of fixed CO_2_ produced by microbes colonizing iron snow aggregates rapidly sinks. Iron snow is characterized by a higher Fe fraction (35%) and lower organic cabron content (11%) compared to marine or lake snow, therefore the estimated iron snow C sedimentation rate (121–600 mg C m^−2^ d^−1^) in Lake 77 is 12× higher than C sedimentation rates in ferruginous Lake Matano (10.56–15.6 mg C m^−2^ d^−1^) [8,9] and 10–20× higher than C sedimentation rates in the North Atlantic (12–30 mg C m^−2^ d^−1^) [106], due to high velocity of iron snow (~2 m h^−1^). Conversely, the C sedimentation rate in Lake 77 is 1–2 orders of magnitude lower than in Lake Constance [8,107]. A previous study showed that anaerobic methanogenesis accounted for more than 50% of organic matter degradation in Lake Matano [9]. Thus, the absence of ^13^C incorporation in heterotrophic FeRB under anoxic conditions in our microcosms suggests the majority of organic carbon was exported to the sediment without significant utilization in Lake 77. The rapid sinking of newly synthesized organic C results in a short residence time and enhanced contribution to the overall carbon pump from the surface to the anoxic sediments in acidic lakes.

Considering both the metatranscriptome and metaproteome profiles, our results show that *Ferrovum* and *Leptospirillum* are responsible for the rapid fixation of CO_2_ coupled to Fe(II) oxidation under oxic conditions and represent the main chemolithoautotrophic members of the community. The diversion of fixed CO_2_ to the biosynthesis of EPS represents the organic carbon fraction of the iron snow aggregates. Furthermore, EPS is subsequently broken down again and likely used for biomass production by *Acidiphilium* sp. and other heterotrophic members of the community. The low labeling in these organisms shows that the EPS pool represents a relatively large pool of organic carbon compared to the new production of EPS, which is only slowly enriched in ^13^CO_2_ by the autotrophs. Instead, other, more readily available and usable carbon compounds released by *Ferrovum* spp. and *Leptospirillum* spp. are used by *Granulicella* spp. for heterotrophic growth, but not by *Acidiphilium* spp., which suggests that *Acidiphilium* spp. is highly specialized to strictly rely on EPS as an organic carbon source and cannot incorporate carbon coming directly from *Ferrovum* and *Leptospirillum* spp.

## 5. Conclusions

FeOB and FeRB showed the highest transcriptional activity for functions linked to carbon fixation, polysaccharide biosynthesis and flagellar motility in the iron snow microbiome. The predominance of FeOB and FeRB in the iron snow aggregates, based on 16S rRNA amplicon sequencing and functional profiling, suggest Fe-cycling bacteria play a pivotal role in iron snow aggregation, stability and overall microbiome structure. Our study shows that chemolithotrophic CO_2_ fixation is the main microbially mediated inorganic C fixation process by the iron snow microbial community, rather than photosynthetic CO_2_ fixation which is the predominant process in marine and lake ecosystems. Dissimilar to oceans and lakes, only a small fraction of the fixed CO_2_ is transferred to the heterotrophic FeRB under oxic conditions, thus indicating limited incorporation of fixed CO_2_ by the heterotrophs in the water column. Thus, iron snow creates a very efficient carbon pump between the surface and the sediment in acidic, Fe-rich ferruginous lakes.

## Figures and Tables

**Figure 1 microorganisms-09-01368-f001:**
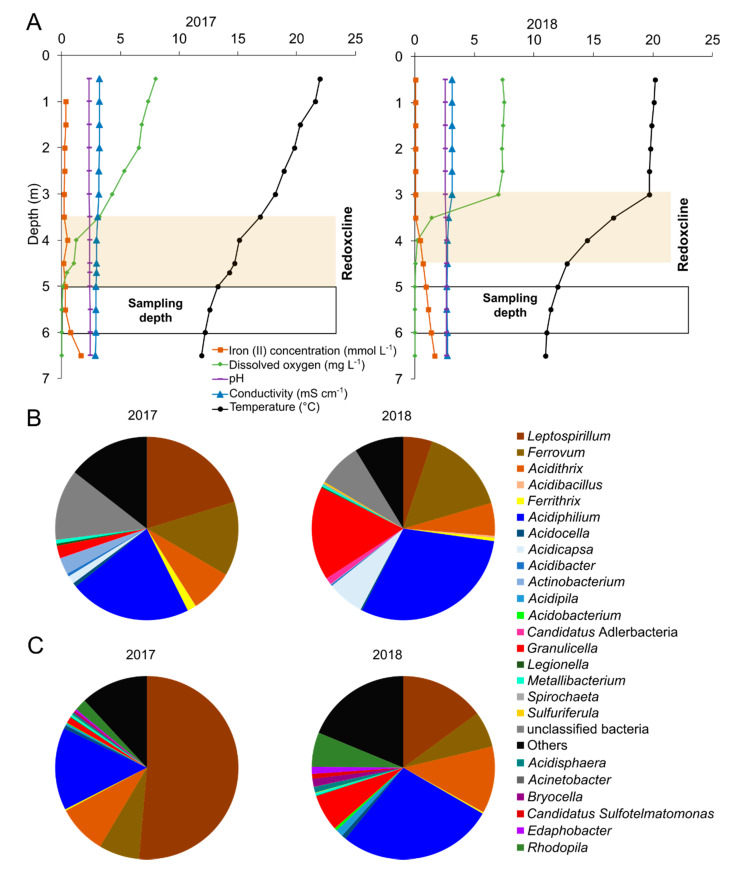
Water profile and microbial taxonomy of iron snow. (**A**) Water profiles of biogeochemical parameters measured at the sampling location in the central basin (CB) of Lake 77 in August 2017 and August 2018. The redoxcline is highlighted in yellow and the range of sampling depth is outlined in black. (**B**) Microbial community composition in iron snow based on 16S rRNA gene amplicon sequencing. Only bacteria classified to the genus level with relative abundance greater than 0.1% are shown. Others represent the sum of bacteria assigned at the genus level, but relative abundance is less than 0.1%. Unclassified bacteria represent bacteria not assigned to any given genus. Data represent the mean of three replicates per year. (**C**) Microbial community composition in iron snow based on mRNA sequences derived from metatranscriptome datasets. Only bacteria classified to genus level with relative abundance greater than 0.5% are plotted.

**Figure 2 microorganisms-09-01368-f002:**
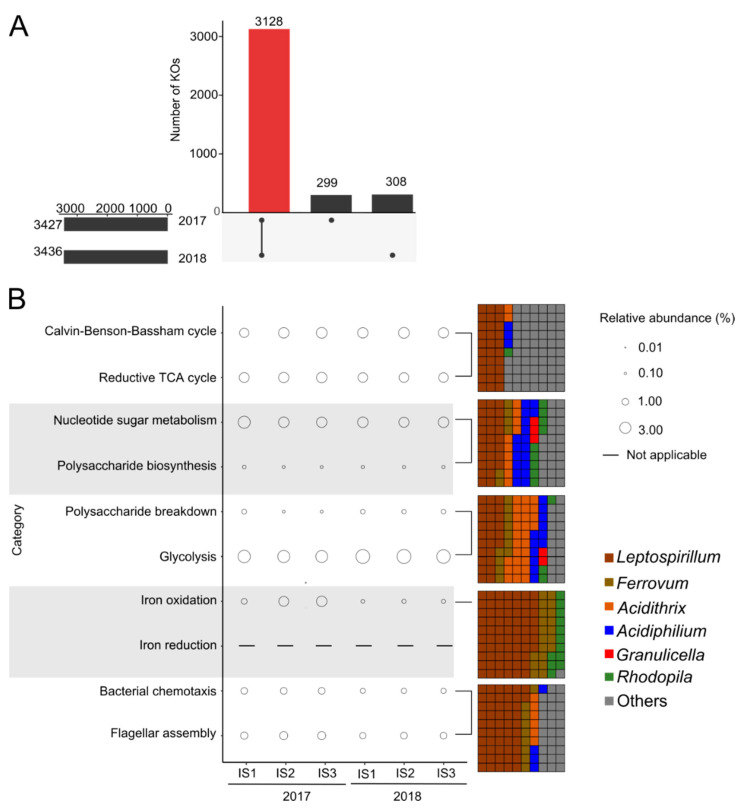
Functional profile of iron snow microbiome in metatranscriptome datasets. (**A**) The UpSetR package [59] was used to visualize shared and specific sets of KOs between the metatranscriptome datasets. Bar charts refer either to the total number of KOs in the datasets or the number of KOs for indicated intersects. (**B**) The relative abundances of functional categories in metatranscriptome datasets. Different colors represent different taxonomies linked to functional categories and only the top 6 taxonomic groups with the highest relative abundance are shown. Waffle charts represent relative abundances of assigned taxonomic groups within respective functional categories of interest. Others represent bacteria associated with one or two functional categories.

**Figure 3 microorganisms-09-01368-f003:**
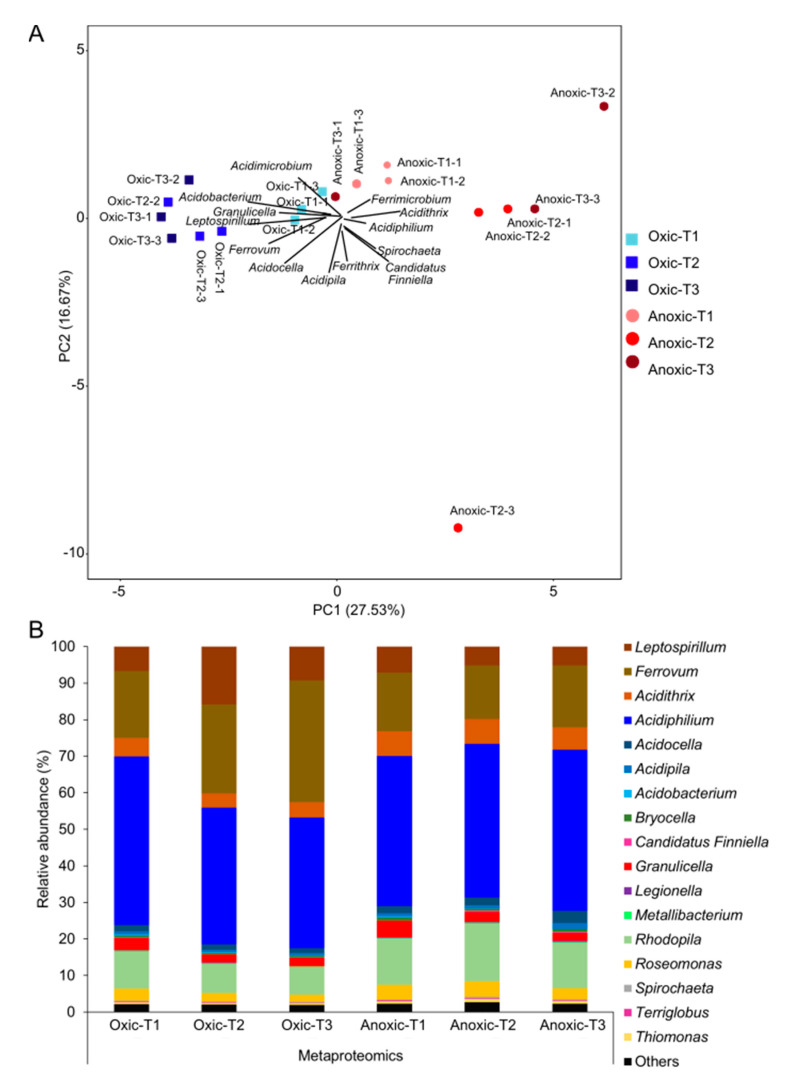
Taxonomic composition of iron snow microcosms in metaproteome datasets. (**A**) Principal component analysis of taxonomic profiles of iron snow microcosms over three time points at the genus level. Triangles represent oxic microcosm incubations and circles represent anoxic microcosm incubations. Different colors represent different time points. T1, T2 and T3 refer to predetermined sampling time points: 7, 14 and 28 days, respectively. (**B**) Microbial community composition based on the relative abundance of peptides of iron snow at the genus level. Only bacteria classified to the genus level with relative abundance greater than 0.1% are plotted.

**Figure 4 microorganisms-09-01368-f004:**
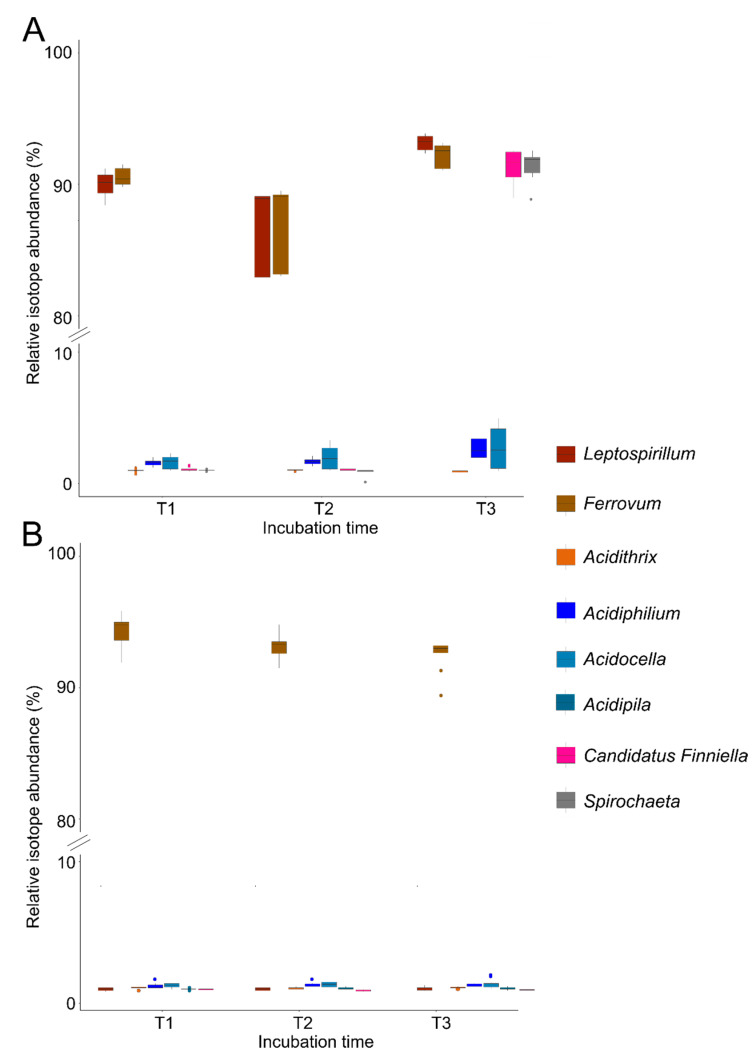
^13^C incorporation patterns of dominant iron cycling bacteria in iron snow microcosms. (**A**) Boxplot of bacterial ^13^C incorporation pattern under oxic conditions. x axis represents three incubation times (T1: 7 days; T2: 14 days; T3: 28 days) and y axis shows the relative isotope abundance (RIA) starting from 0% to 100%. Different colors represent different genera. The dots represent outliers of ^13^C incorporation value away from the median. (**B**) Boxplot of bacterial ^13^C incorporation pattern under anoxic conditions.

**Figure 5 microorganisms-09-01368-f005:**
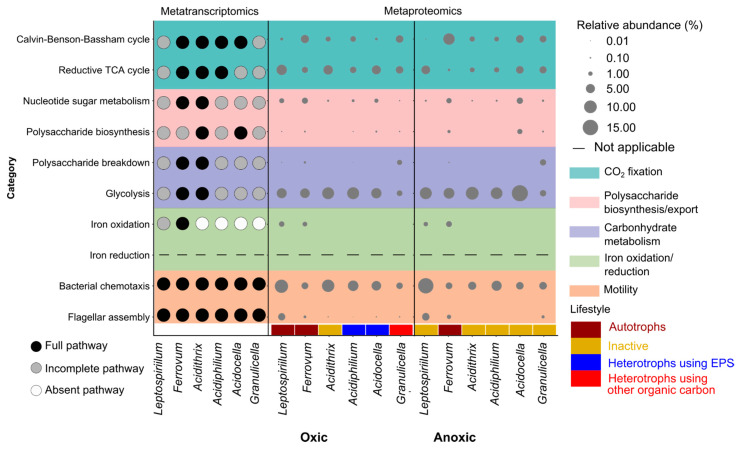
Functional profile of dominant iron cycling bacteria in the metatranscriptome and metaproteome datasets. The left panel represents the transcriptional activity of bacteria based on mapping of mRNA sequences to genomes in 2018 metatranscriptome datasets. The pathway coverage of each genome is shown in Appendix A. Dark, grey, white represent complete, incomplete, absent pathway separately based on mapping coverages of pathways. The right panel represents the proportion of functionally assigned peptides to any KEGG category in bacteria. The missing dot means that the number of peptides affiliated with the category equals to 0. Data represent the mean of replicates at three timepoints.

## Data Availability

The RNA-seq data have been submitted to the ArrayExpress under the accession number E-MTAB-9686. The 16S rRNA Illumina MiSeq amplicon sequencing data of in situ iron snow samples and iron snow microcosms have been deposited at the European Nucleotide Archive EBI-ENA under the accession numbers ERR4690399-4690404 and ERR4690405-4690440, respectively. Metaproteome data have been deposited at PRIDE under accession number PXD025534.

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
