# Peer review of "Insights into Autotrophic Activities and Carbon Flow in Iron-Rich Pelagic Aggregates (Iron Snow)"

_microorganisms, 2021, doi:10.3390/microorganisms9071368_

Round 1

Reviewer 1 Report

Comments on Li et al.’s submission to Microorganisms:

Li and colleagues studied the iron-rich pelagic aggregates (iron snow) in one ferruginous, acidic lake. Specifically, they identified the dominant species of microbes responsible for the primary productivity as iron oxidizers. More interestingly, they reported that most of the fixed C will be deposited to the sediment without further recycling in the water column.

Given the fact that anoxic and ferruginous lakes are closely relevant with the Archean and Proterozoic ocean chemistry, this finding is potentially significant for early Earth studies. The observation of very efficient burial of organic C is novel and may have significant influences on our future understanding of organic C burial in the early ocean(s).

I have no major concerns about their methods and observations. However, I am very curious about why the burial efficiency is so high (even with high sulfate in the seawater), presumably due to the acidic pH(?). Sean Crowe and colleagues did many studies on ferruginous lakes like Lake Matano and didn’t observe such high burial efficiency of organic C. Also, the Archean seawater pH was thought to be weakly acidic to neutral (6.5 to 7; Krissansen-Totton et al., 2018 PNAS; Halevy & Bachan, 2017 Science). If pH is a big factor for organic C burial efficiency, then a short discussion on this topic will certainly improve the implication part of this research.

Minor:

Line 56: reference format is inconsistent.

Author Response

Dear Reviewer,

Thanks for your suggestions regarding the manuscript.

We modified the manuscript and responded to your comments.

Please have a look at the submitted document.

Best,

Qianqian

Reviewer 2 Report

Summary:

Li et. al report a study investigating the microbial community structure and functional capabilities within palagic aggregates (iron snow).  The authors performed 16s rRNA sequencing and metatranscriptomics of microbial communities from an acidic lake below the well oxygenated region of the water column, and found strong evidence for Fe(II) oxidation, carbon fixation and EPS production.  To further link the physiology to community structure, the authors constructed microcosms of these communities under oxic or anoxic conditions, and performed 16s rRNA amplicon sequencing and 13C protein stable isotope probing (protein-SIP) to elucidate which taxa were fixing CO2, and which taxa were directly consuming metabolic byproducts produced by the autotrophs.

Overall, this is a good study. The data generated is valuable for microbial community ecologists, and provides a high resolution of view of chemoautotrophy in acidic, Fe(II)-rich environments, thus warranting publication.  The methods and text are presented in a very clear and organized fashion.  

I have just a few suggestions for additional analysis that could help the readers interpret their results a bit more. 

Comments:

Community structure at different levels of taxonomic resolution.

The authors generated 16s rRNA amplicon sequencing and metatranscriptomics of microbial communities from an acidic lake below the redoxcline.  In Fig.1b, the authors show the community composition at the genus level, and provide relative abundances in Table S3. These values only sum to ~67%, suggesting that a large fraction of ASVs were not classified at the genus level. It would be ideal to compare the community compositions at lower taxonomic resolutions, as this might highlight some additional broad classes of microbes involved in iron cycling.  Furthermore, I am curious to know whether there are some ASVs that are capable of photoautotrophy that are currently undercounted because they cannot be resolved at the genus level.  

Comparison between communities in situ vs. microcosms

The authors performed experiments investigating the community composition and autotrophic capabilities in microcosms in the dark in both oxic and anoxic conditions.  It would be very interesting to look at the 16s amplicon data generated from in situ communities vs. microcosm communities, and report on the differences. This analysis could shed light onto which (if any) taxa are performing photoautotrophy in situ, as these taxa should be expected to decrease in relative abundance in the microcosms. 

Functional characterization of communities

The authors performed metatranscriptomics and proteomics of the communities, and compared the transcripts involved in various chemoautotrophic pathways vs. photoautotrophic pathways.  However, since the taxa known to be capable of photoautotrophy (Rhodopila) is present at low abundance, the metatranscriptomics and proteomics data might miss some key functions encoded by these taxa.  It might be interesting to perform functional imputation of these communities using a standard tool like FAPROTAX (Louca, Parfrey, and Doebeli 2016) or PICRUSt (Langille et al. 2013) to see if any rare functions encoded by low abundant species are worth investigating in future work. 

Oxidants for chemolithoautotrophy

A major result of the paper is that these communities are dominated by Fe(II) oxidizing chemoautotrophs, but it is unclear what the oxidant is driving this metabolism in situ.  There is very low oxygen in the original environment, and the authors speculate that nitrate might be the primary oxidant shaping driving autotrophy in anoxic microcosms.  I am curious to know whether the authors can see signs of nitrate respiration in the anoxic microcosms using either their proteomics data or the imputation of 16s rRNA amplicon data.

Is Acidiphilium photosynthetically active?

In Fig S3, the authors show that Acidiphilium encodes photosynthetic genes, but is often cited as a strict chemoheterotroph.  I was wondering if the authors could comment on this discrepancy in the manuscript.

Proposal for follow up study

Although I believe this is beyond the scope of the current paper, I think it would be very interesting if the authors studied these microcosms in anoxic, illuminated conditions.  From my understanding, theory suggests that Fe(II)-driven photoautotrophy doesn’t work in acidic conditions, and it would be fascinating to see if other electron donors in these acidic environments could be used for photosynthesis. 

References:

Langille, Morgan G. I., Jesse Zaneveld, J. Gregory Caporaso, Daniel McDonald, Dan Knights, Joshua A. Reyes, Jose C. Clemente, et al. 2013. “Predictive Functional Profiling of Microbial Communities Using 16S rRNA Marker Gene Sequences.” Nature Biotechnology 31 (9): 814–21.

Louca, S., L. W. Parfrey, and M. Doebeli. 2016. “Decoupling Function and Taxonomy in the Global Ocean Microbiome.” Science 353 (6305): 1272–77.

Author Response

Dear Reviewer,

Thanks for your comments regarding the manuscript.

We modified the manuscript and responded to your comments.

Please have a look at the submitted documents.

Thank you very much for your great help.

Best regards,

Qianqian
